# A composite metric for evaluating system resilience with non-idealistic performance curves

**Madhura Yeligeti**[1,2]*, **Hans Christian Gils**[1], **Wolfgang Nowak**[2]

1 German Aerospace Center (DLR), Institute of Networked Energy Systems, Stuttgart, Baden-Wuerttemberg, Germany, 2 University of Stuttgart, Institute for Modeling Hydraulic and Environmental Systems, Stuttgart, Baden-Wuerttemberg, Germany

* madhura.yeligeti@dlr.de

## Abstract

Designing systems and processes resilient to sudden shocks is an essential element of system analysis in many engineering fields. Quantitative resilience assessment employs various metrics to examine and monitor system resilience through experimentation. Existing resilience metrics typically portray the system's response to a shock-like event as an inverse bell-shaped, triangular, or trapezoidal curve of performance over time. Then, for example, the downward and upward slopes are interpreted as the disruption and restoration phases of the system, respectively. However, these metrics fail or need simplification when a system response does not exhibit such an idealized shape. In this paper, we introduce a composite metric combining various elements of system performance curves, irrespective of shape features. Additionally, the metric integrates a user-defined critical threshold into its mathematical formulation. To verify the metric's performance, we conducted a survey among researchers in energy system analysis using illustrative system response curves. Comparing the survey-derived ranking and the metric values verifies that the metric aligns with the judgment and expectations of potential users. Finally, we benchmark our metric against its contemporaries, highlighting its versatility with nontypical performance curves. Due to its modular mathematical formulation, this metric can be applied, enhanced, and extended for comparative performance assessment in various fields of analysis, especially in the absence of idealized system response curves.

## 1 Introduction

Since its early formal definition by Holling in 1973 [1], the concept of resilience has evolved across various fields of study. It describes a system's ability to withstand an extreme event, absorb disturbance, restore to an expected steady state, and even undergo transformations to adapt to a new steady state [2]. Here, we associate resilience with the ability of a technical system to maintain its functionality or services under severe stress, with the goal of 'bouncing back' to a steady, stress-free state,

**Data availability statement:** All data required for transparency and reproducibility of this study are provided as supporting information. The images in the supporting information files are available under the Creative Commons Attribution License (CC BY 4.0).

**Funding:** The research in this project was sponsored through the project 'ReMo-Digital' funded by the German Federal Ministry for Economic Affairs and Energy (BMWE) under grant number 03EI1020B, supporting the authors Madhura Yeligeti and Hans Christian Gils. The scientific contributions of Wolfgang Nowak are supported by the Stuttgart Center for Simulation Science (SimTech) The funding parties had no role in the study design, data collection and analysis and the decision to publish, or in preparation of the manuscript.

**Competing interests:** The authors have declared that no competing interests exist.

combining the definitions from Brand et al. [2], Oliver [3] and Folke [4]. From a design and operational perspective, this can be assessed by monitoring the system's performance under disruption. The system response is summarized into a normalized measure of performance (MOP), such that a value of 1.0 signifies a complete or satisfactory performance level. Fig 1 illustrates an MOP-over-time curve from the instant an extreme event strikes ($t_{start}$). Then, the system undergoes disruption and recovery until the end of the event or its effects ($t_{end}$). The exemplified curve marks a classical system performance curve and is at the core of quantitative resilience assessment.

Quantitative methods for resilience assessment in literature have become more complex in recent years, encompassing more parameters, especially domain-specific aspects, to characterize resilience. Despite the large variety of resilience evaluation measures, there are some common traits. While some studies consider a short, extreme shock [5–8], others consider long-lasting effect like climate change [9–14]. In some cases, the event's probability or systems' and components' vulnerability is incorporated in the assessment metrics [15–23]. Some metrics are comprised of several indicators associated with system structure [10,12,14,24,25], system performance [6,7,13,16,17,26–33] or a combination of both [18,20,21,34]. On the contrary, other assessment methods condense systemic performance and design into a single index, or a 'summary metric' [35], as indicated in numerous publications [8,11, 18–20,22,23,36–49]. A few metrics in the literature also include a critical threshold below which system impacts are considered extremely severe, e.g. [5,29,50–52]. An overview of the metrics presented by all the literature mentioned above is available in Appendix A1.

The vast development of resilience metrics and indicators has also been compiled in numerous domain-specific literature reviews, especially for power and energy systems [53–60], supply chains [61,62], and information and communication systems [63,64]. Comprehensive commentaries on general systemic resilience assessment metrics can be found in the work of Cheng et al. [65]. Poulin and Kane [35] offer an excellent overview of the different types of summary metrics of infrastructure resilience and the performance measures used to derive them. They emphasize that performance measures can represent system availability, system productivity, or service quality. The type of performance measures required to compute the resilience metric is determined by the function, scope, and field of assessment of the system. A recent publication [66] presented a compilation of resilience indicators in the form of a library for the *Python* programming community with modular, predefined functions for metrics from the fields of bio-science, network science and information theory.

Advanced metrics in recent literature aim to extract the maximum possible information from the system's performance curve to better represent or quantify resilience. They typically define three phases of system response: disruption, recovery, and adaptation. Consequently, these metrics include elements such as slope/rapidity of failure and recovery, disruption time, recovery time, etc. c.f. [6,17,19, 22–25,27,28,33,38,42–44,46,50,67,68]. This categorization of system response relies on the presumption of a typical triangular, trapezoidal, or, in colloquial terms, 'bathtub' shape of the performance curve (cf. Fig 1). However, all such metrics become incompatible when such an idealized shape of the performance measure does not

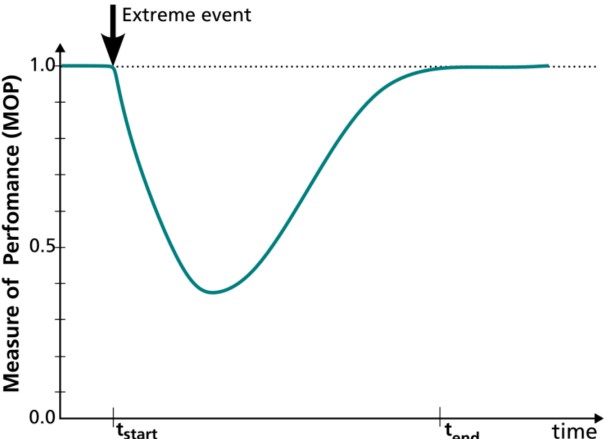

**Fig 1**. **Typical system performance curve on impact.**

apply. In complex systems, when several interacting components undergo individual damages and recoveries following stress, the resulting MOP curve over time can appear strange in shape. In such cases, an approach for decomposing the curve to identify failure and recovery parts of performance may be required to apply known metrics. This is indicated in [27] where real outage data from the transmission network is used to derive the system performance curves. Another example of real-world system complexities leading to atypical performance curves is presented by Silva et al. [69] in context of employment payroll index during multiple recessions in US. Atypical performance curves are also common in networked systems that deal with dynamic demand-supply relationships. Performance measures typically selected for these systems represent resource availability or supply adequacy, which can haphazardly vary based on localized impacts. The following example illustrates this.

In the field of energy system analysis, various optimization frameworks like REMix [70], oemof [71], and PyPSA [72] are used to design and analyze future energy systems at a national to international scale. Capacity adequacy lies at the core of the optimization problem, and the prime constraint is to ensure enough supply through generation and transport to meet energy demand throughout the system's scope. Hence, the typically chosen performance measure is the amount of energy supplied, expressed as a fraction of energy demand. When some part of such infrastructure fails due to a sudden extreme event, the optimization framework reorganizes the operation of the energy system. It transfers energy from unaffected areas to affected regions, trying to meet demand as much as possible. This reorganization is limited by the available and functional infrastructure. However, during this time, both energy demand and availability may also vary over time, e.g. due to daily variations in demand or due to fluctuating solar or wind resource availability. Thus, the resulting MOP curve does not always form the typical shape as in Fig 1. Instead, the resulting curves show somewhat atypical shapes, as in Fig 2, where no apparent disruption or restoration phases can be identified.

Such atypical curves are not explicitly addressed in the literature. Here, the resilience metrics rely on single parameters derivable from the curve, independent of the shape. Examples are the overall area under the performance curve, the total duration of performance loss, or the difference to a performance threshold. These components are used individually, i.e. the metric reflects only one aspect and is not composed of several components. Evidently, these metrics are used in literature for complex, networked systems such as water supply systems [33], drainage systems [73], electrical networks [29], communication systems [40] and transportation systems [52] also summarized in a review by Zhou et al. [74]. As an alternative, some resilience metrics used in literature are not based on the system's performance but derived from system topology, design, or other properties [16,24,25,30,41,49]. However, these metrics represent system design and structure, not necessarily the system's response to stressful events.

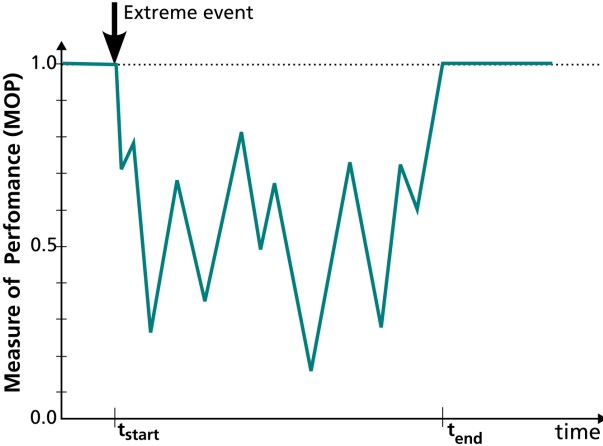

**Fig 2**. **Example of system performance curve without explicit failure and recovery phases.**

In summary, our literature review shows that advanced performance-driven resilience metrics are mainly based on idealized shape aspects of performance curves (slope, time, disruption and restoration phases, etc.). When such a standard curve is absent, an overall performance measure like area or duration is simply interpreted as the metric, or the metric is formulated completely independent of performance. Additionally, in all the studies we reviewed, the metrics are designed with a focus on a specific field of application. Hence, the value-added by the metric is only demonstrated via implementation in a domain-specific case-study. Even among studies presenting metrics that can be extended to other fields in principle, a domain-independent verification and benchmarking of the metric is missing.

This paper addresses the above-mentioned gaps through three contributions. Firstly, we propose a composite, performance-driven resilience metric that synthesizes insights from existing literature while integrating multiple aspects of system performance into a single summary measure, independent of curve shape. The metric is designed with modular, intuitive mathematical functions, allowing both broad applicability and flexible fine-tuning. Secondly, we verify the robustness of this formulation through a survey using synthetic, atypical performance curves. Thirdly, we benchmark our metric against established metrics from literature, in capturing non-idealistic system behavior and highlight its applicability in the landscape of quantitative resilience assessment.

## 2 Materials and methods

This section describes the design process of the metric including the requirements (Sect 2.1) and the mathematical formulation (Sect 2.2). The concept and assessment of the survey are explained in Sects 2.3 and 2.5, respectively. The metrics used for benchmarking are introduced in Sect 2.6. All the evaluations are presented in Sect 3.

### 2.1 Requirements of the metric

As indicated above, this summary metric is aimed at performance-based resilience assessment. It should be concise yet capture the essence of the system's shock response. It should be derivable simply from the 'measure of performance' (*MOP*) curve over time (e.g. Figs 1 and 2). The performance measure itself can be defined as suitable to the system type or the study domain. In the following, further expectations lay out foundational assumptions.

The metric should undoubtedly indicate the overall performance loss, generally represented by the area $A$ between the performance curve and the standard performance (cf. Bruneau et al. 2003 [75]). This area is the integral of performance loss (1–*MOP*) over time, over a time period of evaluation $T_{eval}$ that runs from $t_{start}$ to $t_{end}$ (cf. Figs 1 and 2).

This area should be weighted more significantly if the same performance loss is observed for a less intense event. Hence, at least one event-based characteristic, like duration or intensity, should be a part of the metric's equation. This will enable a fair comparison of metric values under the effect of different events.

Since there is no standard unit for the metric of resilience, we deliberately define this metric as unit-free and limited to a finite range with fixed definitions for maximum and minimum values. This requirement is built on the prerequisite of a normalized, unit-less measure of performance ($0 <= MOP <= 1$). Being unit-free can allow broader applicability and usability of metrics. As the bounding range, we consider the interval of [0,1], where the metric takes the value of '0' only when the system does not recover to full operation long after the event effects have receded. The value of '1', on the other hand, indicates no drop in performance during and after the extreme event. The two extreme cases are unique, and the metric value should lie between '0' and '1' for all the other cases. This includes cases when the system goes through a complete shutdown momentarily or even longer. As long as the system recovers back to the adequate performance measure, the metric is non-zero, unlike a few other metrics in literature, e.g. [68]. Keeping an intuitive metric range enables different metric values to be compared against one another and assessed with respect to the best and worst performance.

The metric should also be influenced by the minimum value ($MOP_{min}$) of the measure of performance ($MOP$) during the time of evaluation $T_{eval}$, being especially penalized if $MOP_{min}$ is close to zero, i.e. complete shutdown. Practically, a complete shutdown of system infrastructure is rare. Most systems and processes try to maintain a minimum performance level of some components essential for operating critical or fall-back infrastructure. In large systems and processes, this may represent one or more assets and facilities like water, energy, health, security services, transportation, etc. Since we assess only the integrated system performance, these individual requirements can be combined into a 'critical' value, $MOP_{crit}$, of the measure of performance. The value of $MOP_{crit}$ can be interpreted as the performance level below which the damage is more severe since even the most critical services fail. Such instances indicate even lower resilience and the metric values should reflect this severity. This 'critical' value, $MOP_{crit}$, is an additional parameter determined by the system analyst's discretion. By default, however, $MOP_{crit}$ can simply be considered zero.

The eventual mathematical formulation of the metric should obey all the above requirements and be derivable for both continuous and discrete time steps and all shapes of a *normalized* performance curve.

## 2.2 Mathematical formulation

From Sect 2.1, we can summarize four major factors derivable from the performance curve that should be incorporated in the metric:

1. Area ($A$), between the MOP curve and perfect performance measure over an evaluation period $T_{eval}$
2. The minimum value occurring for $MOP$ ($MOP_{min}$)
3. Duration that $MOP$ lies below $MOP_{crit}$, named as $T_{crit}$
4. Indicator of complete recovery at the end of the evaluation period, $R_{rec}$

These factors are used in individual Eqs (2)–(5) to construct the four principal components, namely $R_{area}$, $R_{min}$, $R_{crit}$ and $R_{rec}$ that constitute the overall metric $R$ as shown in Eq (6). Since $R \in [0, 1]$, $R_{area}, R_{min}, R_{crit}$ and $R_{rec}$ should also have a maximum of 1 and minimum of 0. Note that the time coordinate in the metric formulations is denoted with the letter $t$ in lower case, while parameters and variables indicating time duration are named with an upper case $T$.

Two extrinsic, user-defined parameters are involved in the metric formulation. One of them is the critical value for the measure of performance, namely $MOP_{crit}$ as defined earlier. The other is the evaluation time, $T_{eval}$, which fulfills the requirement of an event-based characteristic in the metric. $T_{eval}$ represents the duration of either the event itself or its effects on the system. For complex systems, the event's timing and duration may differ from the timing and duration of its impact on the system. For instance, in the event of a storm, an electricity transmission infrastructure, like power lines and poles, may face damages gradually, and restoration works will surely continue even after the storm. Hence, $T_{eval} =$

$t_{end} - t_{start}$ is defined by the user by choosing the point in time $t_{end}$ where one expects the system to have recovered fully, considering the type of event and the particular system in perspective. $t_{start}$ is given by the onset of the system response to the disruptive event. This user-defined parameter $T_{eval}$ thus represents the period over which the system response is evaluated. It is not to be confused with the duration of the event that causes the disruption, although it is influenced by the event. A consistent $T_{eval}$ should be used for metric calculations when comparing multiple system responses to the same extreme event. When different events need to be accounted for, the different $T_{eval}$ values play a role in fair resilience estimation.

Fig 3 illustrates all the components constituting the metric in a performance-curve graphic.

Now, let us proceed with the four components one by one. The first component $R_{area}$ focuses on the overall loss in performance represented by the area $A$ between the performance curve and the measure of adequate performance $MOP = 1$, as explained in Eq (1):

$$A = \int_{t_{start}}^{t_{end}} (1 - MOP(t))\, dt \tag{1}$$

If the area is zero (no loss in performance), $R_{area} = 1$. From there, the value of $R_{area}$ should reduce as this area increases, indicating an inverse relationship. However, the time frame of evaluation $T_{eval}$ should scale the impact of $A$. The significance of $A$ should be higher if the same loss in performance occurs for a smaller event. The resulting formula for $R_{area}$ is presented in Eq (2).

$$R_{area} = e^{-A/T_{eval}} \tag{2}$$

The ratio $A/T_{eval}$ can be interpreted as the arithmetic mean performance loss over the evaluation time frame. In principle, as $A/T_{eval} \to \infty$, $R_{area} \to 0$. However, due to the definition of $A$, the largest possible value for $A/T_{eval}$ is 1, so that $R_{area} \in [1/e, 1]$. The non-zero lower limit satisfies our choice that resilience is zero if, and only if, there is no complete recovery in the evaluation period, which we handle with a separate component of our metric.

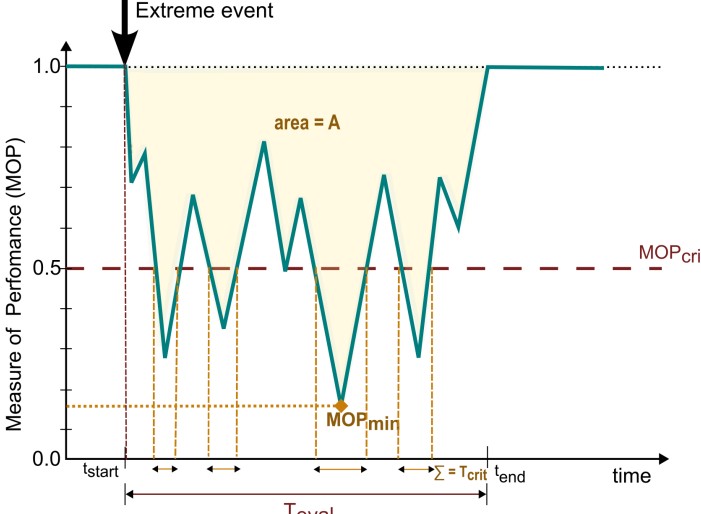

**Fig 3**. Illustration of metric components in an exemplary non-idealized system performance curve.

The next component, $R_{min}$, considers the minimum measure of performance $MOP_{min}$ encountered throughout the evaluation period. The lower the performance, the higher the extent of system dysfunction, even if it is instantaneous. Suppose two systems show the same average loss of performance, i.e. $A/T_{eval}$. One of them shows an extreme loss of performance over a shorter time, while the other one has a moderate loss over a longer time. The former case is deemed more critical, particularly if the performance drops below $MOP_{crit}$. To engineer the corresponding equation to complement $R_{area}$, we have three considerations:

1. Clearly, $R_{min} = 1$ occurs when $MOP_{min} = 1.0$, indicating no performance loss. Then, the $R_{min}$ should decrease as $MOP_{min}$ decreases.
2. The severity of $MOP_{min}$ should differ based on the critical performance level $MOP_{crit}$. If $MOP_{min} > MOP_{crit}$, the overall stress response is satisfactorily above critical values. So, $R_{min}$ should remain close to one, indicating only a little penalty. This should drastically change around $MOP_{min} = MOP_{crit}$, even more as the gap $MOP_{crit} - MOP_{min}$ increases.
3. The effect of $R_{min}$ should not be too drastic. Specifically, it should not reach zero even at zero $MOP_{min}$, as zero is reserved solely for the case of non-recovery.

To satisfy the above requirements, $R_{min}$ should be either a logarithmic function or a fractional polynomial. The term $MOP_{min}{}^{MOP_{crit}}$ fits perfectly to cause the shift in slope at $MOP_{min} = MOP_{crit}$ while ensuring values between zero and one. To host considerations one and three, we linearly re-scale this expression, arriving at the Eq (3), with $R_{min} \in [0.5, 1]$.

$$R_{min} = 0.5 \cdot \left( MOP_{min}{}^{MOP_{crit}} + 1 \right) \tag{3}$$

The third component of the proposed metric, $R_{crit}$, highlights the duration $T_{crit}$ for which $MOP$ stays below $MOP_{crit}$. We choose to normalize $T_{crit}$ by $T_{eval}$ to indicate not the time but the time fraction of evaluation where $MOP < MOP_{crit}$.

To design the equation for $R_{crit}$, we demand that $R_{crit} = 1.0$ if $T_{crit} = 0$. From there, $R_{crit}$ should reduce as $T_{crit}/T_{eval}$ increases. When $T_{crit}/T_{eval} = 1$, indicating that all time steps of the curve are below $MOP_{crit}$, $R_{crit}$ should attain its minimum. However, it should still not entirely drop to zero since, again, the only zero condition in the metric is incomplete recovery until $t_{end}$. These conditions can very well be represented by a negative exponential curve as presented in Eq (4):

$$R_{crit} = exp \left( -\frac{T_{crit}}{T_{eval} \cdot (1 + MOP_{crit})} \right) \tag{4}$$

Now, it remains to reason for the additional appearance of the expression $(1 + MOP_{crit})$. If the user-specified $MOP_{crit}$ value is larger, there is a higher chance for $MOP$ to drop into the critical range, i.e. below $MOP_{crit}$. To reflect this fact, $MOP_{crit}$ should be integrated into the metric, albeit weakly, such that if the same $T_{crit}$ is obtained for a low value of $MOP_{crit}$, it is slightly worse than if it were occurring for a high value of $MOP_{crit}$. We achieve this by including $MOP_{crit}$ in the denominator within the exponential. Adding a positive constant number (here chosen as 1.0) to $MOP_{crit}$ ensures $R_{crit}$ exists even if the assumed $MOP_{crit}$ is zero and weakens the influence of $MOP_{crit}$ as desired. The resulting interval for $R_{crit}$ is $[1/e, 1]$.

The last component is $R_{rec}$. Each of the above three components is curated to lie in a particular range, such that full resilience in the respective aspect is signified by values of 1. By definition, the overall resilience metric should be zero only if the recovery is incomplete by the end of the user-defined time window, i.e. $MOP < 1$ at $t_{end}$. Hence, $R_{rec}$ is a binary variable to indicate whether $MOP$ at $t_{end}$ is one ($R_{rec} = 1$) or not ($R_{rec} = 0$):

$$R_{rec} = \begin{cases} 0, & \text{if } MOP \text{ at } t_{end} \neq 1.0 \\ 1, & \text{otherwise} \end{cases} \tag{5}$$

This also makes the $R = 0$ condition unique from all other cases independent of $R_{area}$, $R_{min}$ and $R_{crit}$.

Finally, the metric value is generated by superimposing the effects of the four factors. Therefore, the four terms $R_{area}$, $R_{min}$, $R_{crit}$ and $R_{rec}$ are combined by multiplication as shown in Eq (6).

$$R = R_{area} \cdot R_{min} \cdot R_{crit} \cdot R_{rec} \tag{6}$$

It can be argued that another way to combine the four components could be a (weighted) sum, which is linear and computationally more straightforward. However, the key feature of our choice of product is that low values of single terms greatly influence the overall value. That means a single weak aspect among the four components cannot be compensated easily by the others. In the most extreme case, a single zero makes the overall result zero. A comparison of this approach with a weighted sum is presented for some sample performance curves in Appendix A2.

### 2.3 Concept for the survey

The purpose of the metric is to provide a concise quantitative basis to compare system performances after stressful events based on their performance curve. To verify that the proposed metric quantifies performance loss for resilience studies in an intended way, we performed a survey comparing exemplary performance curves. The survey aims to check whether the metric delivers a ranking in alignment with the visual expectation of scientists with reasonable experience in either energy system analysis or resilience assessment. In addition, the sample performance curves used in the survey can also verify if the calculated metric values are well-distributed within the possible range of [0,1] for diverse performance curves.

For the survey, we designed eight performance curves denoted as $c_1 - c_8$. They are illustrated in Fig 4 and the corresponding MOP values are provided in the supporting information S1 File. The extrinsic parameters $T_{eval}$ and $MOP_{crit}$ are common for all the curves. $t_{start}$ and $t_{end}$ are also consistent across all eight curves.

During the survey, each question showed two curves at a time, and the respondents were asked to select the least resilient option between the two. Eight curves, combined two at a time, yield 28 combinations, forming 28 questions. To keep the survey size small enough to promote complete responses, the 28 questions were split into two questionnaires of 14 each. The two questionnaires were distributed in the scientific research community in direct contact with the authors of this study. The two questionnaires were randomly distributed within the group, such that each person received only one questionnaire. This scientific community comprised of:

1. Scientists involved in the consortium of the project ReMo-Digital funded by the former German Federal Ministry for Economic Affairs and Climate Action (BMWK), which focuses on the resilience of energy systems
2. Scientific researchers at the German Aerospace Center's Institute of Networked Energy Systems (DLR-VE)

For transparency, both survey questionnaires are provided as *supplementary material* available as S2 File and S3 File. In the survey, the participants were first informed about the research context, objectives, and purpose of the survey before presenting the questions.

### 2.4 Ethics statement

The survey did not contain any mandatory questions, and participants could skip any question as they wished. Thus, every step of the survey participation, from opening the questionnaire, answering the questions to submitting the responses has been completely voluntary and anonymous. Hence, the action of participation itself was considered as consent to process the survey responses for scientific research, and no other form of consent was deemed necessary.

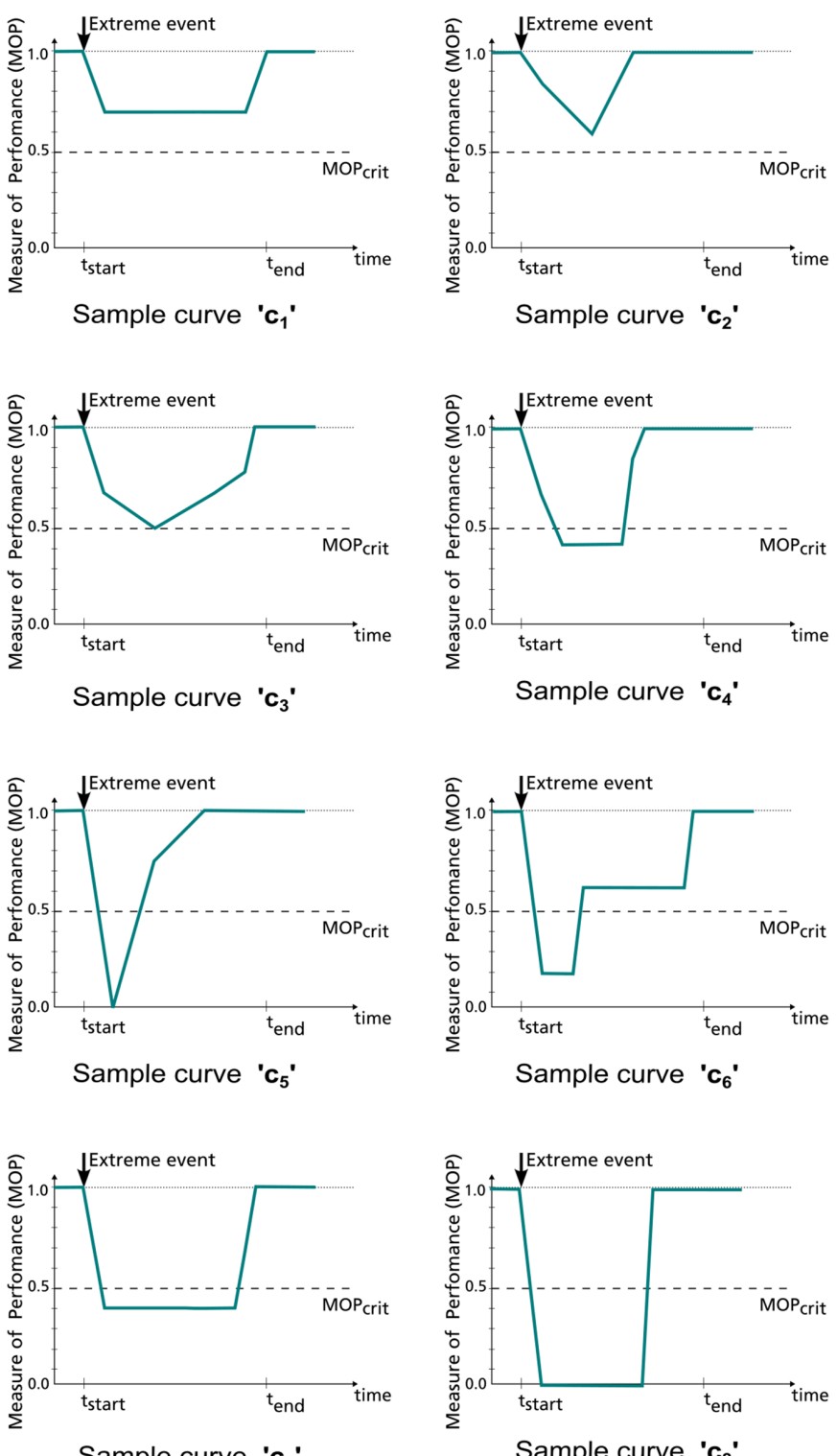

**Fig 4**. **Example system performance curves to extreme event response generated for the survey.**

Since, human participation in the survey was limited to providing opinions on synthetic, exemplary resilience curves that do not relate to any ethics-sensitive issues, the authors did not seek an approval from an ethics committee for distributing the survey and collecting responses. However, at the time of submission of this study in January 2025, the 'Office of Ethics in Research' at the German Aerospace Center (DLR e.V.) was consulted. On reviewing the survey questionnaires and the process of the survey, they waived the need for an ethical review retrospectively in written form.

The survey was distributed in early February 2023 and it was closed by the end of the same month. Therefore, the survey-based assessment is based entirely on the responses collected during this period. No other archived or documented data was collected or processed in the framework of the survey. The survey did not collect any personal information, and all responses were completely anonymous. Consequently, it is not possible to trace a response back to the participant. The collected data is provided in original form as supporting information S4 File and S5 File. This data was then analyzed to generate a ranking, as explained in Sect 2.5.1.

## 2.5 Assessment of survey responses

**2.5.1 Pairwise comparison for ranking.**  When compiling all the responses, we calculated a pairwise comparison matrix $X$. It compares all eight curves in all possible combinations so that matrix element $x_{ij}$ refers to the comparison of curve $c_i$ against curve $c_j$, with $i, j = 1 \dots 8$. Individual scores count the number of votes that curve $c_i$ received in comparison against curve $c_j$. These votes are responses to questions appearing in either of the questionnaires. To account for the discrepancy between the number of responses in survey questionnaires 1 and 2, each score is divided by the number of responses for the respective question it comes from. Thus, all terms in the matrix lie between 0 and 1, the diagonal elements remain empty, and, by principle, diagonally opposite values add to 1.0. Results can be found in Sect 3.1.

Applying various pairwise ranking methods in voting theory can generate a rank chronology among all the curves from the pairwise comparison matrix. We adopt two versions of the pairwise-ranking method, both finding origins in the works of Ramon Llull in the 13th century [76]. While Llull proposed methods to find simply the winner from pairwise voting, the development of a ranking strategy can be attributed to Nikolaus Cusanus (15th century) and Jean-Charles de Borda (18th century) [77].

Applying the *Cusanus-Borda* method [76] to pairwise voting, the scores across all the columns for each curve $c_i$, i.e. each row of the pairwise-comparison matrix, are added up. The resulting scores, arranged in descending order, correspond to the resilience-based ranking of the sample curves.

The second approach is a more recent development or reinvention of *Llull's* method, acknowledged as the *Copeland* method based on Arthur Copeland's lectures in 1951 [78]. In this method, the pairwise comparison matrix undergoes a transformation. All the values lying above 0.5 are marked as 1.0. All the values lying below 0.5 are marked 0.0. All the values lying at 0.5 are marked 0.5. The next step is to add these marked values across the columns for each row just like the *Cusanus-Borda* method. The resulting scores, arranged in descending order, represent the rank.

While the first method accounts for the number of votes received in each pairwise comparison to determine the rank, the second method focuses only on the number of pairwise trials won. The two methods can potentially derive different rank orders [76]; therefore, both are used for metric verification. All the scores and resulting ranks from the survey are presented in Sect 3.1.

**2.5.2 Calculation of confidence intervals.**  Each of the above ranking methods condenses all the responses into a single rank chronology, without providing any information on how confident one can be about the ranking, e.g. if the survey group had been much larger. Instead, suppose the survey could be repeated many times with a broad range of survey-takers. In that case, one can obtain

1. A large set of possible rank chronologies of the eight sample performance curves, indicating ranges of ranks or even probabilities of how often a curve would take a rank
2. A confidence interval for the rank of each curve.

Since, in practice, the experiment cannot be repeated several times, we resort to resampling with bootstrapping. Bootstrapping was introduced and expanded by Bradly Efron [79,80] in the late 20ᵗʰ century. It is used to draw statistical inferences for problems with a limited sample size (e.g., a small number of survey participants) by repeated resampling with replacement. By adding information about confidence, bootstrapping creates a robust basis for our metric verification. We apply bootstrapping directly at the level of survey participants. By resampling from the survey participants in each of the two questionnaires, each repetition of resampling generates a new, randomized set of survey results. From this, we can re-compute the pairwise comparison matrix and obtain a new, randomized ranking result according to Sect 2.5.1. By repeating this many times, we can approximate the probability that the survey-derived ranking matches the metric-derived ranking.

Over these bootstrapped repetitions, we keep everything else the same. That means if questionnaire 'A' (14 questions) originally received '$n_A$' unique responses, and questionnaire 'B' (the other 14 questions) received '$n_B$' unique responses, we draw '$n_A$' participant response sets (with replacement) from the responses to questionnaire $A$ and '$n_B$' participant response sets from questionnaire $B$. Each 'response set' is a complete set of answers actually provided by a participant. Thus, the consistency of a survey-taker's response is maintained, and the resampling does not mix different people's answers into one response.

This resampling exercise is repeated many times, until sufficient statistical convergence of the results (here: the computed rank probabilities) is achieved. For our purposes, we define convergence as sufficient and terminate the procedure when the computed rank probabilities stay consistent within a tolerance of 0.0001 over at least 20 consecutive iterations. To better evaluate convergence, and for robustness of probabilities and confidence intervals, we repeated the entire bootstrapping process five times, see Sect 3.1.

## 2.6 Benchmarking against existing metrics

While recent performance-based metrics in literature cannot be applied to every atypical performance curve (e.g. Fig 2), a few metrics, with some assumptions, may be applied to the sample performance curves (cf. Fig 4) since their shapes are similar to the standard triangle or trapezoid. This offers a possibility of comparing the proposed metric with other metrics used in systemic resilience assessment. Our primary selection criteria for other 'single-valued' summary metrics is that the metric can be calculated from normalized system performance curves without additional information. The second condition is that it should be possible to apply the selected metric irrespective of the specific research domain it comes from and that it can be used for all eight sample curves with limited assumptions and approximations. For a competitive comparison, metrics considering at least two features of the performance curve instead of simply area $A$ or minimum measure of performance $MOP_{min}$ are preferred. Most summary metrics in literature (also summarized in Appendix A1), meeting these criteria, are also influenced by the shape characteristics of the performance curve. Yet, we select three metrics to illustrate possible metric rankings for the sample performance curves. The metrics selected are the ones proposed by Yarveisy et al. [42], Cheng et al. [43], and Najarian et al. [44]. An advantage of this selection of metrics is that they all share a fixed range [0,1] that matches the range of our proposed metric. The mathematical equations of these metrics are presented in Appendix A3 along with the assumptions made to apply them to the eight performance curves. The resulting rankings from the different metrics are compared in Sect 3.2.

## 3 Results and discussion

In this section, we present and discuss the results of the survey assessment. The metric-generated ranking is viewed in comparison to the confidence interval generated by bootstrapping the survey responses (Sect 3.1). This is followed by benchmarking results (Sect 3.2). Finally, we discuss the limitations of the metric and highlight the scope for future work (Sect 3.3).

## 3.1 Resilience ranking of the sample performance curves

Before the survey results are assessed, the mathematical formulae presented in Sect 2.2 are applied to each of the curves (cf. Fig 4), and the resulting '$R$' values are presented in Fig 5. In principle, the higher the metric values are, the higher the resilience.

The ranks for the curves based on these values are then compared with survey-generated ranks. The survey questionnaires 1 and 2 received 22 and 19 distinct responses, respectively. As mentioned earlier, the collected responses are available as supporting information (S4 File and S5 File). During the survey, the survey-takers were asked to rate their expertise in 'Resilience' and 'Energy System Analysis' on a scale of 1 to 5, where '1' indicates a negligible and '5' a very high level of expertise. Fig 6 and Fig 7 illustrate the expertise distribution among the survey-takers in the two fields. Since energy system analysis is the key application of the metric in our research, the survey was distributed particularly to the energy system analysis community. Therefore, the question also catered to expertise in energy systems. This expertise helps us to judge how well the responses represent the expectations of potential resilience researchers in the energy system analysis community. Expertise in resilience assessment is a bonus since knowledge of different methods and metrics adds an edge to the judgment of performance measures. While most survey takers did not consider themselves resilience experts, they were fairly confident about their expertise in energy system analysis, as seen in the two plots. When added together, the scores yield an average of 5.5 over 10, which is considered sufficient for assessing the simplistic performance curves in the survey.

The pairwise-comparison matrix is presented in Fig 8. Each term, say $x_{ij}$ in the pairwise comparison matrix, is generated by adding all the responses that claim $c_i$ shows less resilience than $c_j$, divided by the total responses to the question of $c_i$ v/s $c_j$. Since the survey takers were asked *'What is worse?'*, higher scores represent lower resilience. The figure shows that the results are almost unanimous for curve samples lying at the extremes, e.g. $c_1, c_2$ and $c_7, c_8$. The poll gets competitive in the middle of the matrix. Especially for the comparison of $c_5$ and $c_6$, the votes are split equally.

The scores generated by the Cusanus-Borda and Copeland methods explained in Sect 2.5.1 are presented in Table 1. Again here, the higher the score, the lower the resilience. For easier visual comparison, the scores from both scoring methods are inverted in Figs 9 and 10. Since each curve is compared to 7 other curves, the maximum possible score for any curve is 7.0. Hence, all scores are subtracted from 7.0 for the presentation of ranks. In principle, the absolute values of both methods' calculated and inverted scores carry no significance individually. However, for the Cusanus-Borda method, the strong or weak majority in the survey responses is reflected in the score values.

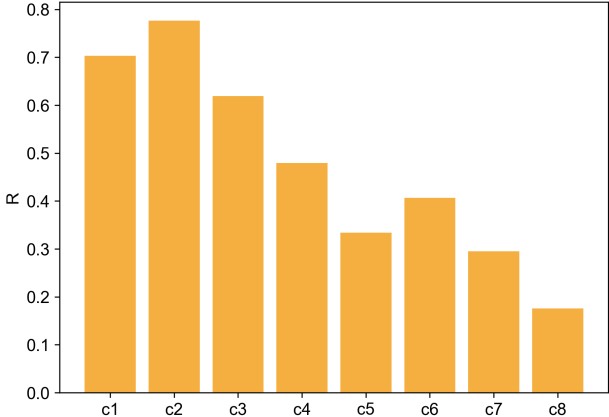

**Fig 5**. Resilience metric 'R' values for the sample performance curves.

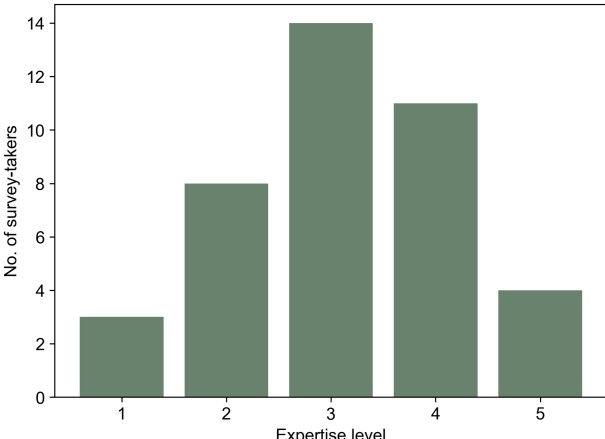

**Fig 6**. Distribution of expertise among survey takers in the field of energy system analysis.

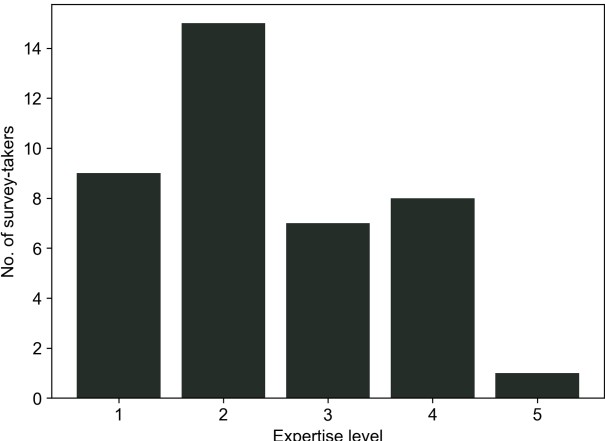

**Fig 7**. Distribution of expertise among survey takers in the field of resilience assessment.

As seen from the figures, the ranking generated by our metric tallies almost perfectly with the rank chronology from survey responses according to both ranking methods. The only exception is the tie between $c5$ and $c6$ indicated by the Copeland method since both received equal votes when compared against each other (also seen by the '0.5' in the pairwise comparison matrix). According to the Cusanus-Borda method, the difference in the scores of $c_5$ and $c_6$ is only marginal, indicating that the ranks could have been switched easily if only a few opinions had changed.

Ranking without bootstrapping assigns a probability of 1.0 for the calculated rank chronology. Without bootstrapping, our metric ranking perfectly matches the ranking of the Cusanus-Borda method. Thus, the probability that the metric rank matches the survey-derived rank is 1.0. Fig 11 shows how the computed probability changes (and converges) throughout 5,000 bootstrapping iterations for five overall repetitions. These are the probabilities that a repeated survey would result again in the same ranking as the original survey, exactly matching the metric-generated ranking. For each of the five overall repetitions, convergence is achieved according to our predefined criterion at about 3,000-4,000 iterations. Hence, we finally select a safe count of 5,000 iterations to estimate the desired probabilities and confidence intervals. At 5,000 iterations, the probability of the metric-matching rank chronology is within the range of 0.55 to 0.58 for all five

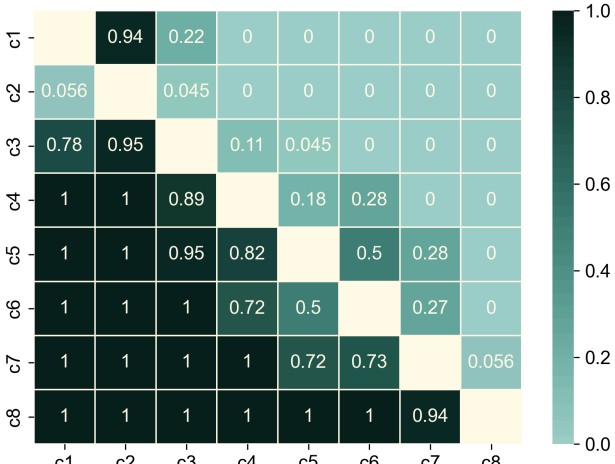

**Fig 8**. Color-map representation of the pairwise-comparison matrix depicting normalized scores of performance curves' comparison from survey responses; diagonal values are null.

**Table 1**. Scores for each curve calculated with the Llull and Copeland scoring systems using the pairwise-comparison matrix.

| Curves | $c_1$ | $c_2$ | $c_3$ | $c_4$ | $c_5$ | $c_6$ | $c_7$ | $c_8$ |
|---|---|---|---|---|---|---|---|---|
| Cusanus-Borda scores | 1.17 | 0.10 | 1.89 | 3.35 | 4.55 | 4.49 | 5.51 | 6.94 |
| Copeland scores | 1.0 | 0.0 | 2.0 | 3.0 | 4.5 | 4.5 | 6.0 | 7.0 |

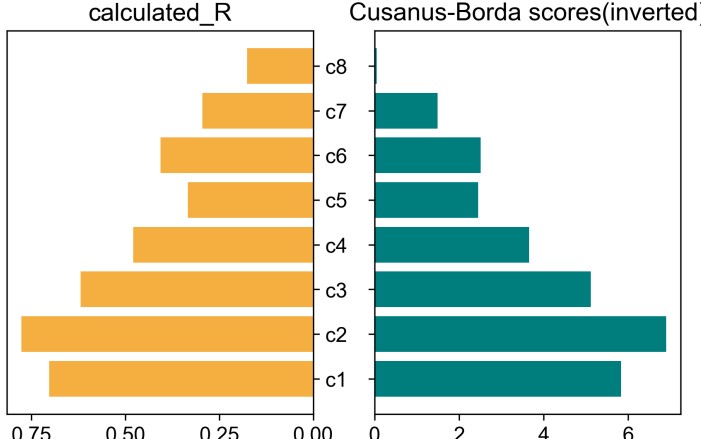

**Fig 9**. Rank comparison of calculated 'R' with survey ranking by 'Cusanus-Borda' method.

repetitions. Hence, we infer that if the survey experiment is repeated several times, the probability that the Cusanus-Borda ranking of Fig 9 will be achieved is about 55-58 %. This is interpreted as the probability that the metric ranking matches the survey-derived ranking in a robust experimental setup.

It is also important to note which rank chronologies constitute the other 42-45 % probability. Within 5,000 iterations, five other rank chronologies occur. However, a substantial share of the overall probability (about 40%) is taken by a rank chronology very similar to the one achieved by the metric values. The only difference is the swap between '$c_5$' and '$c_6$'.

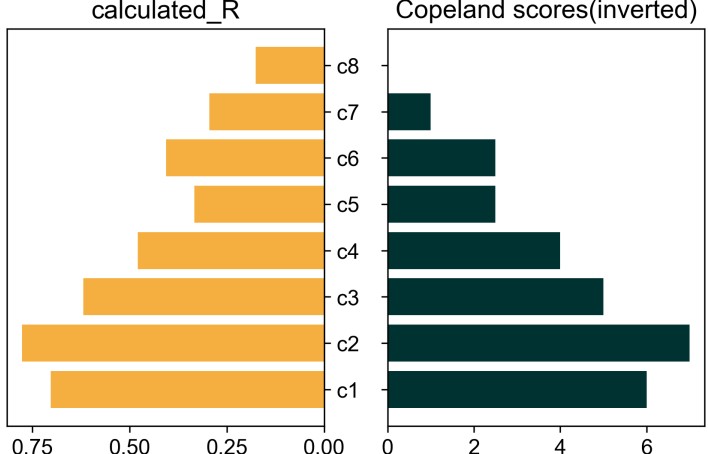

**Fig 10**. **Rank comparison of calculated 'R' with survey ranking by 'Copeland' method.**

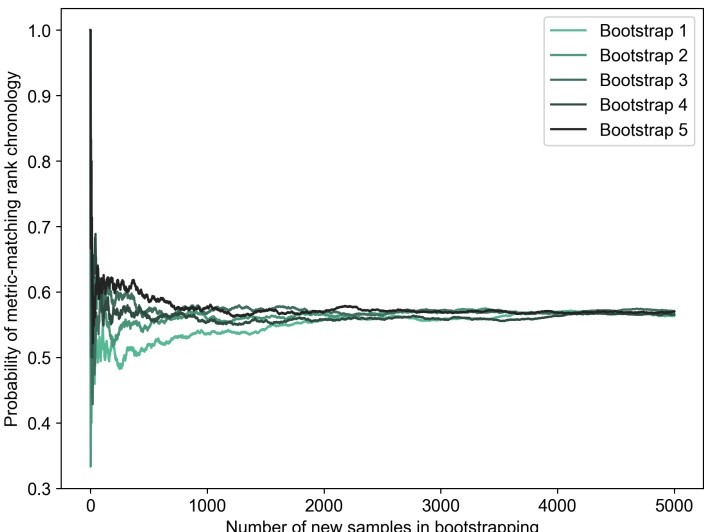

**Fig 11**. **Convergence of probability of metric-matching ranking for five bootstrapping repetitions.**

This swapping aligns with high variance in responses around $c5$ and $c6$ as seen in the pairwise comparison matrix of Fig 8. All six rank chronologies occurring in bootstrapping and their probabilities to occur are presented in Fig 12.

When repeating the bootstrapping even further toward $N \rightarrow \infty$, theoretically, all possible rank orders will appear. But their probabilities will be insignificant. About 99 % of the probability will still be encompassed by the two major rank chronologies, as with $N = 5000$. These two rank chronologies thereby determine the confidence intervals of the possible ranks for each of the eight curve samples. As illustrated in Fig 13, the 99 % confidence intervals for the curves $c1$–$c4$ and $c7$–$c8$ are fixed to the same ranks as those obtained even with the metric. The only spread is seen in the intervals of $c_5$ and $c_6$; even here, the metric ranking perfectly matches the interval's median. The narrow range of the confidence intervals indicates a high unanimity in most of the responses, which can also be noted in the pairwise comparison matrix. The confidence intervals mark a boundary for acceptable ranks for each sample curve.

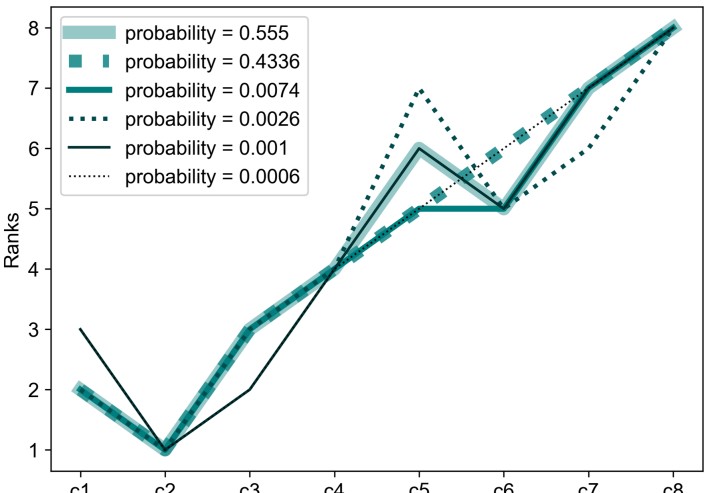

**Fig 12**. All rank chronologies and their respective probabilities in 5000 iterations of bootstrapping, illustrated for 1 of the five repetitions.

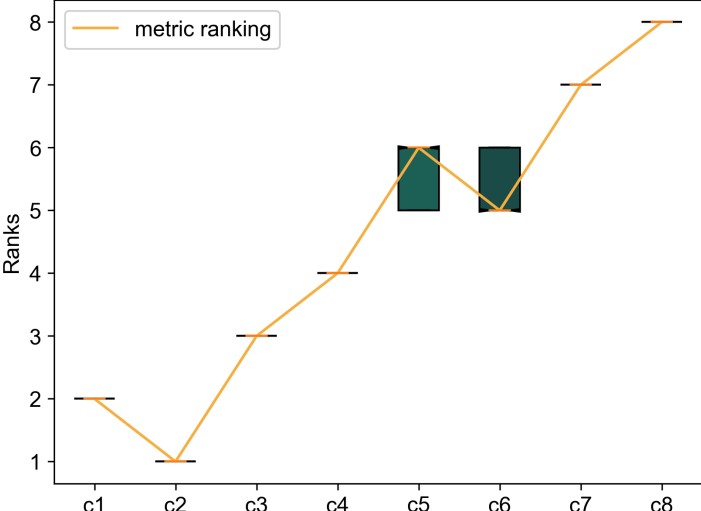

**Fig 13**. Rank comparison of calculated 'R' with confidence intervals of survey-based ranks calculated with the Cusanus-Borda scoring method.

With bootstrapping-based verification, we can overcome the bias due to small number of responses. However, since the survey was distributed mainly in the energy system analysis community, there may also be field-specific biases in their assessment of the performance curves although the sample curves were designed to be domain-independent. This bias cannot be removed unless the survey experiment is undertaken at a much broader scale. Since that is not feasible, we acknowledge the bias and complement survey-based verification with benchmarking against other metrics.

### 3.2 Benchmarking

For the comparison with our metric proposed, $R$ values for the performance curves from Fig 4 are calculated with the other metrics selected in Sect 2.6. The assumptions made for the required weights and other parameters for the corresponding metric formulations are specified in Appendix A3. The resulting ranking of the performance curves is then compared to that of the proposed metric (Fig 14).

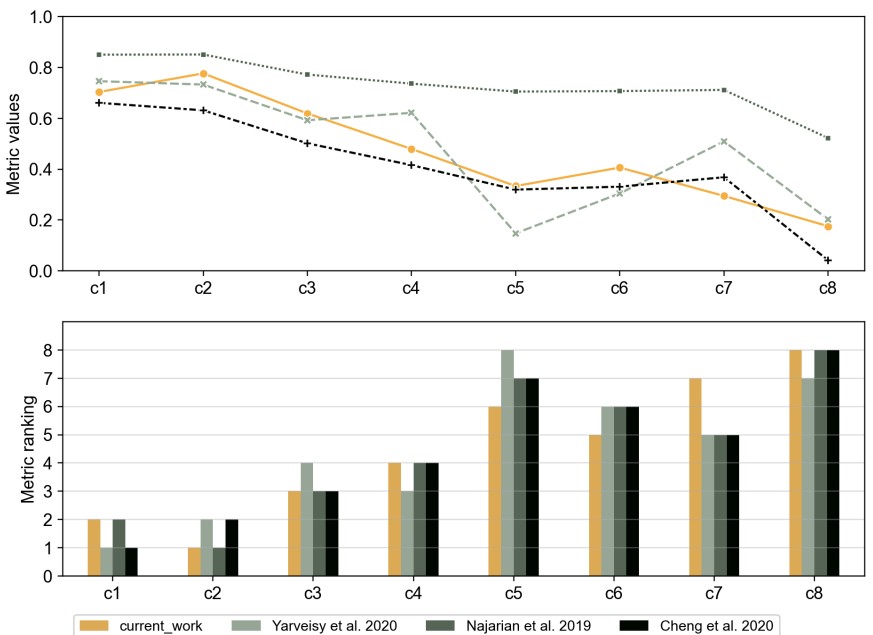

**Fig 14.** **Comparison of metric values and resulting ranks of the proposed metrics and the three selected metrics for benchmarking.**

Evidently, none of the metrics' rankings exactly match each other. However, all the metrics show a common ranking trend, especially in the selection of the top ranks. The consistently ranked bottom four sample curves, namely $c_5 - c_8$, indicate much higher drops in performance. This is the zone where the specific characteristics of each metric dominate the ranking. Two peculiar examples can illustrate this.

While most metrics indicate $c_8$ as the worst performing curve, just like the unanimous vote for $c_8$ as the worst in the survey, the metric by Yarveisy et al. [42] ranks it above $c_5$ in resilience. This can be explained by the following. The metric by Yarveisy et al. [42] considers the specific trapezoidal curve shape with smooth disruption and recovery stages and a steady adaptive stage in between. Accordingly, for the non-trapezoidal curve shapes $c_2 - c_6$ (cf. Fig 4), the metric formulation is applied assuming the path until minimum MOP as disruption and remaining path until full recovery as restoration. This metric allocates the highest importance to the absorptive capacity (captured by the minimum MOP here). Thus, $c_5$ and $c_8$ with $MOP_{min} = 0$ have drastically low resilience values compared to other sample performance curves since $MOP_{min} = 0$, indicating zero absorptive capacity. Between $c_8$ and $c_5$, $c_8$ seems to show an adaptive phase and rapid recovery compared to $c_5$'s lack of an adaptive phase (non-trapezoidal curve shape) and comparatively more recovery time. Thus, $c_8$ outweighs in performance than $c_5$, according to the metric by Yarveisy et al. [42].

Secondly, all three benchmarking metrics rank $c_6$ and $c_7$ opposite to the proposed metric. For the metric by Yarveisy et al., this can be explained by the heavy influence of the absorptive capacity in the metric formulation. The lower drop in performance of $c_7$ acts as a major advantage for $c_7$. Secondly, the metric cannot exactly capture the shape of $c_6$. Hence, the path from the last time step where $MOP = MOP_{min}$ is considered as the restorative phase (approximating it to a straight line, for more information, see Appendix A3). Thus, $c_6$ is seen as having a short adaptive and long restorative period, compared to $c_7$, leading to a lower resilience value for $c_6$. The metrics by Najarian et al. [44] and Cheng et al. [43] do not explicitly specify a shape. Still, they assume a typical disruption-like drop in performance, followed by a rise in performance called restoration. For the atypical sample curves, the disruption time is considered as the time to reach minimum MOP, and everything after it is considered recovery or adaptation. The metric components of Najarian et al.

integrate the MOP values over the disruption and restorative phases. The metric's third component represents the disruption and restoration duration, which is the same for $c_6$ and $c_7$ with given assumptions. Hence, the determining factors are absolute MOP values during disruption and restoration. The weights for all components are simply considered equal. Thus, a relatively large weight is given to the few time steps when the disruption occurs, where $c_6$ clearly is worse than $c_7$. This causes the overall metric value to favor $c_7$ as more resilient over $c_6$ even though the overall MOP values or areas indicate otherwise. Similarly, equal weights are assumed for absorptive and restorative capacity and reference time while applying the metric by Cheng et al. [43]. Due to an approximation again to simply two phases, the trend of metric values is very similar to that obtained while using the metric by Najarian et al. [44]. This metric's overall values are lower since the disruption component is weighted by the disruption performance measure (here $MOP_{min}$). The focus here is on the average performance measures in the disruptive and restorative phases. Hence, the metric value for $c_7$ is higher than the neighboring $c_5$ and $c_6$.

Each metric used for comparison involves an elaborate mathematical formulation aiming at capturing several aspects of the performance curve. All three metrics focus on disruption and restoration behaviors. However, the differences in the relative importance of each phase and the parameters used cause discrepancies in the rankings. Mapping the disruption, recovery, and adaptation phases on all eight performance curves is only possible with approximations that alter the actual loss in performance for some curves. In other words, the metrics can not be applied perfectly to all atypical performance curves. Besides, neither of the three metrics considers a critical threshold, which is a determining factor for ranking with our proposed metric.

### 3.3 Application of the metric

The metric formulation incorporates two user-defined parameters, $MOP_{crit}$ and $T_{eval}$, that influence the metric values and should be applied judiciously. For example, when the overall performance is above the critical range, then $R_{crit} = 1.0$, raising the metric value. A very low $MOP_{crit}$ may lead to high metric values for most performance curves and narrow the distinction among them. The metric is designed such that $R = 0$ occurs only when the MOP does not converge to 1.0 until the end of the evaluation time frame. Hence, the resilience measure, by definition, will drop to the trivial case of zero for many instances if $T_{eval}$ is set to very short period lengths. Similarly, if $T_{eval}$ is much higher than the disruption and recovery time of most performance curves, the metric values will be similarly high for all such curves. These factors must be considered when applying the metric.

Essentially, the scope of the metric is limited to a quantitative assessment of systemic resilience driven by a normalized performance measure. Even though the metric measures the 'resilience' of the system, not all elements that constitute resilience are captured effectively in a single 'measure of performance'. Additionally, the absolute metric value itself carries no substantial meaning except when compared to the best and worst possible values, i.e. $R = 1.0$ and $R = 0$, respectively, or when compared to another value. Therefore, the metric cannot be interpreted as a holistic measure of resilience. However, given the need to investigate system resilience with numerous stress tests, the metric provides a concise and condensed quantitative basis for comparative assessment.

Moreover, the metric's modular nature allows easy tuning and extension to suit the application's scope and needs. Incorporating a time-varying critical threshold that is higher or lower based on the need for system functionality or services is one instance to improve metric application. Similarly, adding other performance components for e.g. the frequency of fluctuations around the average performance loss, can enhance the metric formulation, provided their possible correlation to already used components is acknowledged.

### 4 Conclusion

Quantitative resilience assessment in literature is fueled by metrics and indicators designed to capture the design of systems and their responses to external extreme events. The proposed resilience assessment metric addresses the domain

of performance-driven metrics and tries to contrast the heavy reliance on shape characteristics like rapidity of failure and recovery. The metric caters mainly to non-idealized system responses, facilitating quantitative resilience assessment of complex, interconnected systems (e.g. transport networks, supply chains) where a typical triangular or trapezoidal performance curve does not occur.

We verify the metric by comparing its resilience ranking on eight example performance curves with survey-derived rankings, showing consistency with user expectations. Although the metric verification includes survey-takers mainly from the community for energy systems' research, the metric formulation is independent of field of application. It is relevant for all systems where a normalized performance measure can be determined. This normalized measure can be, for instance, the supply of commodities with respect to demand in supply chains or the number of successful journeys with respect to planned trips in transport systems.

An additional metric verification is benchmarking against other metrics in literature. This yields two main insights. Firstly, it reveals the loss of information when non-idealized response curves are forced into conventional disruption–restoration phases, thus highlighting the applicability of our metric. Secondly, it showcases how different foci in metric formulations produce diverse resilience rankings. This underscores the importance of understanding which aspects of system performance a metric emphasizes before applying it in resilience assessment.

Overall, the proposed metric enables a large-scale comparative resilience assessment by condensing the essence of system performance into a single value. As a disadvantage, it overlooks intricate details of system response, which may be relevant for a comprehensive analysis. In such cases, the metric acts as a perfect entrée for identifying and filtering critical cases before in-depth investigations.

## Acknowledgments

The authors thank the survey takers from the scientific community for their anonymous participation and feedback.

## Appendix

### Appendix A1.  Overview of metrics in literature

For the literature review, the scientific search tool 'Web of Science' from Clarivate™ was used with the search query 'resilience' 'AND' 'metric' to appear in either the title, keywords or the abstract. The resulting 304 results have been short-listed to 46 studies that consider a service or performance-oriented definition of resilience, focusing on system response to extreme event(s). Emphasis is given to mentioning recent publications here since, in most cases, they build upon existing metric formulations, thus encompassing ideas from older publications. Table 2 presents a summary of these resilience metrics.

### Appendix A2.  Metric: Sum v/s Product

This section shows a quick comparison of the metric values calculated by the proposed method of the product of $R_{area}$, $R_{min}$, $R_{crit}$ against their weighted sum for the eight sample performance curves (see Fig 15). $R_{rec}$ is ignored, presuming $R_{rec}$ =1 for all curves. The weights are taken equal for all three components, i.e. 0.33. The range of values achieved by the product is wider than that of the 'sum', emphasizing that the distinction among system performances is more pronounced when the product is used instead of the sum, thus making comparative assessment clearer.

### Appendix A3.  Metrics for Benchmarking

In the following, the parameters and equations of the three metrics used for benchmarking, and the translations and assumptions made while applying the metrics to the eight sample performance curves are described.

**Resilience metric by Yarveisy et al. [42]** is built of three elements to measure the absorptive, restorative, and adaptive capacities, respectively. It considers explicitly a trapezoidal reliability vs time curve with smooth disruption and recovery stages. This reliability is interpreted as the measure of performance (MOP) for our calculations. According to Yarveisy et al., [42], the absorptive capacity, i.e. the system's ability to maintain high residual performance, is represented by the

**Table 2**. Summary of studies presenting event-based resilience metrics.

| Reference | Year of Publication | Type of extreme event(s)[a] | Type of Metric(s)[b] | Dependency on performance curve[c] | Ease of extendability[d] |
|---|---|---|---|---|---|
| Hu et al. [50] | 2025 | S | B2 | 2 | 0 |
| Sapkota et al. [15] | 2025 | S | B2 | 1 | 1 |
| Sajwan et al. [36] | 2025 | S | C2 | 0 | 0 |
| Laino et al. [5] | 2025 | S | B2 | 0 | 0 |
| Abantao et al. [26] | 2024 | S | B1 | 1 | 0 |
| Dobson et al. [27] | 2024 | S | B1 | 1 | 0 |
| Sanabria-Fernandez et al. [9] | 2024 | L | A2 | 0 | 0 |
| Toumasis et al. [37] | 2024 | S | B2 | 0 | 0 |
| Chen et al. [31] | 2023 | S | B1 | 2 | 2 |
| Perri et al. [10] | 2023 | L | A1 | 0 | 0 |
| Roth et al. [11] | 2023 | L | A2 | 0 | 2 |
| Malek et al. [6] | 2023 | S | B1 | 2 | 1 |
| Clemente et al. [12] | 2023 | XL | A1 | 0 | 0 |
| F. Tolner et al. [13] | 2023 | XL | B1 | 0 | 0 |
| Waller et al. [7] | 2023 | S | B1 | 0 | 0 |
| Yao et al. [8] | 2023 | S | B2 | 2 | 2 |
| Dehghani et al. [28] | 2023 | S | B1 | 2 | 2 |
| Gui et al. [30] | 2023 | S | B1 | 0 | 0 |
| Flores-Larsen et al. [16] | 2023 | S | B1 | 0 | 0 |
| Pagano et al. [24] | 2022 | S | A1 | 0 | 2 |
| Rosales-Asensio et al. [38] | 2022 | S | B2 | 2 | 2 |
| Xie et al. [17] | 2022 | S | B1 | 2 | 2 |
| Hutchkinson et al. [29] | 2022 | S | B1 | 0 | 1 |
| Kandeperumal et al. [18] | 2022 | S | C2 | 2 | 1 |
| Raoufi et al. [51] | 2021 | S | B2 | 1 | 2 |
| Yang et al. [19] | 2021 | S | B2 | 2 | 2 |
| Mathew et al. [39] | 2021 | S | B2 | 1 | 1 |
| Liu et al. [14] | 2021 | XL | A1 | 0 | 0 |
| Caskey et al. [25] | 2021 | S | A1 | 0 | 2 |
| Barbeau et al. [40] | 2021 | S | A2 | 0 | 0 |
| Fattahi et al. [41] | 2020 | S | B2 | 0 | 1 |
| Yarveisy et al. [42] | 2020 | S | B2 | 2 | 2 |
| Cheng et al. [43] | 2020 | S | B2 | 2 | 2 |
| Behzadi et al. [32] | 2020 | S | B1 | 1 | 1 |
| Najarian et al. [44] | 2019 | S | B2 | 2 | 2 |
| Assad et al. [20] | 2019 | S | C2 | 0 | 1 |
| Dubaniowski et al. [45] | 2019 | S | B2 | 0 | 0 |
| Jain et al. [21] | 2018 | S | C1 | 0 | 0 |
| Roach et al. [33] | 2018 | S | B1 | 2 | 2 |
| Cai et al. [46] | 2018 | S | B2 | 2 | 2 |
| Hossain-Mckenzie et al. [47] | 2018 | S | B2 | 1 | 1 |
| Novak et al. [34] | 2016 | S | C1 | 0 | 1 |
| Song et al. [48] | 2016 | S | A2 | 0 | 1 |
| Ayyub et al. [22] | 2014 | S | B2 | 2 | 2 |
| Francis et al. [23] | 2013 | S | B2 | 2 | 2 |
| Rosenkrantz et al. [49] | 2009 | S | A2 | 0 | 2 |

[a] sudden, short event: **S**, long term event (e.g. climate change impact): **L**, short events in a long term period: **XL**.

[b] type of systemic indicators used (structural: **A**, performance: **B**, both: **C**); metric represented by several indicators: **1**, condensed to a single value: **2**.

[c] no dependency on perform ance curve: **0**, metric based on the curve but independent of curve slope or shape: **1**, metric based heavily on the performance curve including slope, disruption and recovery phase duration etc.: **2**.

[d] metric is domain- or system-specific: **0**, applicable to multiple fields only with significant changes: **1**, directly applicable to several domains: **2**.

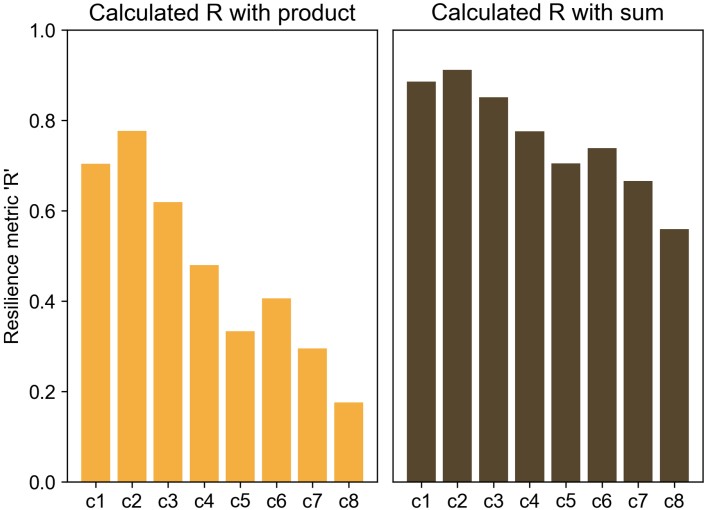

**Fig 15**. **Comparison of metric values calculated by product and sum of the three components** $R_{area}$, $R_{min}$ **and** $R_{crit}$.

drop between the pre-event performance and the maximum performance drop in the trapezoidal performance curve. The steady system operation in the disruptive state is understood as adaptive capacity and recovery (the other edge of the trapezoid) is termed restorative capacity.

For the non-trapezoidal curve shapes $c_2 - c_6$ (cf. Fig 4), the metric formulation is applied assuming the path until minimum MOP as disruption and remaining path until full recovery as restoration. The adaptive phase only exists for the sample curves if the system performance idles at the minimum MOP (e.g. $c_1$, $c_4$, $c_6$, $c_7$, $c_8$). Note that this essentially approximates the shapes of the sample curves as a trapezoidal or triangular shape, modifying the apparent recovery and disruption phases of the performance curve into straight lines. The metric formulations attuned to the nomenclature of this study and assumptions for other variables and constants are described in the following equations.

$$RE = Ab + Ad * Res - Ab * Ad * Res \tag{7}$$
$$Ab = (MOP_{disruption} * C_{ab}/MOP_{max})$$
$$Res = \frac{\arctan \left[ \dfrac{\dfrac{MOP_{max} - MOP_{disruption}}{t_{end} - t_{restoration}}}{t_{end} - t_{start}} \right]}{90} * C_T * C_R$$
$$Ad = 1 - \left( \frac{t_{restoration} - t_{disruption}}{t_{end} - t_{start}} \right)$$

here,

1. $MOP_{disruption} = MOP_{min}$.
2. $MOP_{max} = 1.0$ for all sample curves.
3. $C_{ab} = 1$ since it considers the effect of aging on overall performance and is not considered for the sample curves.
4. $t_{restoration}$ represents the time step at which recovery starts, i.e. this is the last time step where $MOP = MOP_{min}$.
5. $t_{disruption}$ represents the time step when max. disruption hits, i.e. the first time step where $MOP = MOP_{min}$.

6. $C_R$ denotes the drop in performance post-recovery with respect to target performance. Since, post-recovery, all curves maintain $MOP = 1.0$, there is no drop, hence $C_R = 1.0$.
7. $C_T = (t_{restoration} - t_{start})/(t_{end} - t_{start})$, representing the share of time when recovery has not begun after the event.

**Resilience Metric (RM) by Najarian et al. [44]** is also composed of three components similar to that by Yarveisy et al. [42], representing absorption ($r_1$), adaptation ($r_2$) and time-to-recovery ($r_3$). However, unlike the former metric, this metric does not explicitly consider the trapezoidal shape of the curve. Instead, it follows through the disruption and recovery phases, assuming a smoother (inverse-bell-shaped) curve. The metric identifies that the system may recover to a steady state different than its initial state. Hence, it includes an additional parameter for 'target performance' equal to its pre-event performance. In this study, it is termed as $MOP_{target}$, and the time required to reach this stage since the event is termed as $T$, equivalent to $T_{eval}$ of this study. Since the metric's performance drop starts at $t = 0$, the terms $t = 0$ and $T$ are translated to $t_{start}$ and $t_{end}$ respectively. When the term $T$ is intended as time duration instead of a time step, it is replaced with $T_{eval}$.

The metric formulation is a linear combination of all the three components. It is described by the following equations; the applicable assumptions taken for the sample performance curves are also presented below.

$$r = \lambda_1 * r_1 + \lambda_2 * r_2 + \lambda_3 * r_3 \tag{8}$$

$$r1 = \frac{\int_{t_{start}}^{t_{disruption}} MOP(t)dt}{\int_{t_{start}}^{t_{disruption}} MOP_{target}dt}$$

$$r2 = \frac{\int_{t_{disruption}}^{t_{end}} MOP(t)dt}{\int_{t_{disruption}}^{t_{end}} MOP_{target}dt}$$

$$r3 = 1 \ \ if \ T_{eval} <= T_0$$
$$= T_0/T_{eval} \ \ otherwise$$

here,

1. $\lambda_1, \lambda_2, \lambda_3$ are positive weights summing to 1.0. For the benchmarking, all weights are assumed equal ($\lambda_1 = \lambda_2 = \lambda_3 = 0.33$) for all the performance curves.
2. $t_{disruption}$ represents the time step when max disruption hits, i.e. the first time step where $MOP = MOP_{min}$.
3. $MOP_{target}$ is the desired MOP, i.e. before the advent of disruption. In all the curves, $MOP_{target} = MOP_{max} = 1.0$.
4. $T_0$ is a user-defined parameter indicating the standard overall time since the event begins for the system to recover. For all sample performance curves in this study, we take $T_0 = T_{eval}$.

**Integrated resilience metric by Cheng et al. [43]** is also built in a modular manner. Still, unlike the previous two, it consists of two components, one representing absorption or disruption and the second one representing restoration. The system performance from the beginning of the disruption event until it hits its minimum is the disruptive phase, and the rise of the performance from the minimum back to the steady state is the recovery phase. Like the previous study, this study also assumes the typical performance curve shape, except that here, it is not trapezoidal but triangular, similar to Fig 1. Both phases are characterized by three factors in the metric. These are the process factor $\delta$ (following the MOP values through the phases), the consequence factor $\sigma$ (representing the MOP value reached at the end of the phase with respect to the start of the phase), and the time factor $\rho$ (indicating the duration of the phase). These factors and the met-

ric formulation are described in the following, with the values assumed for applying the metric to the sample performance curves.

$$R_l = \alpha * \delta_d * \sigma_d * \rho_d + \beta * \delta_r * \sigma_r * \rho_r \tag{9}$$

$$\delta_d = \frac{\int_{t_{start}}^{t_{disruption}} MOP(t)dt}{(t_{disruption} - t_{start}) * MOP_{start}}$$

$$\sigma_d = MOP_{min}/MOP_{start}$$

$$\rho_d = \Delta^{\frac{B}{(t_{disruption}-t_{start})}}$$

$$\delta_r = \frac{\int_{t_{disruption}}^{t_{end}} MOP(t)dt}{(t_{end} - t_{disruption}) * MOP_{start}}$$

$$\sigma_r = MOP_{end}/MOP_{start}$$

$$\rho_r = \Delta^{\frac{(t_{end}-t_{disruption})}{B}}$$

here,

1. $\alpha$ and $\beta$ are weights associated with the disruptive and restorative phases, respectively. $\alpha, \beta \in [0,1]$ and $\alpha + \beta = 1$. Since there is no preference for any phase for our study, we set both weights equal, i.e. $\alpha = \beta = 0.5$.
2. $t_{disruption}$ represents the time step when max disruption hits, i.e. the first time step where $MOP = MOP_{min}$.
3. $MOP_{start}$ represents the pre-event steady-state measure of performance. For the sample performance curves, this is 1.0.
4. $t_{start}$ refers to the time the event strikes.
5. $\Delta$ is the degradation factor managing the relative importance of time in the equation. It is prescribed that $0 < \Delta \leq 1$. So, the higher the $\Delta$, the higher the significance of the time aspect. Since there is no particular significance or lack thereof intended for the sample performance curves, $\Delta = 1.0$ is taken for benchmarking.
6. $B$ is a reference unit of time indicating a baseline in hours or days, etc., based on system requirements and balancing the absolute time unit of the duration for each phase, maintaining the overall metric unit-free. In this study, the reference time is set $B = T_{eval}$, which is equal for all the sample curves.
7. $t_{end}$ refers to the time when the system recovers to a steady state, i.e. the end of the evaluation time frame.
8. $MOP_{end}$ is the steady MOP achieved at $t_{end}$. The metric allows $MOP_{end}$ to be different from $MOP_{start}$. But in the case of the sample performance curves, $MOP_{start} = MOP_{end} = 1.0$.

## Supporting information

**S1 File. MOP values for sample performance curves.** This file contains the MOP v/s time values considered for the eight sample performance curves in Fig 4.
(CSV)

**S2 File. Survey Questionnaire 1.** This file contains one questionnaire of the survey distributed in February 2023. The questionnaire was set up using the Forms software provided by Google®.
(PDF)

**S3 File. Survey Questionnaire 2.** This file contains another questionnaire of the survey distributed in February 2023. The questionnaire was set up using the Forms software provided by Google®.
(PDF)

**S4 File. Responses to survey questionnaire 1.** This file contains the responses to the questionnaire in S2 File, as collected in original form.
(CSV)

**S5 File. Responses to survey questionnaire 2.** This file contains the responses to the questionnaire in S3 File, as collected in original form.
(CSV)

## Author contributions

**Conceptualization:** Madhura Yeligeti, Wolfgang Nowak.

**Data curation:** Madhura Yeligeti.

**Formal analysis:** Madhura Yeligeti.

**Funding acquisition:** Hans Christian Gils.

**Investigation:** Madhura Yeligeti, Hans Christian Gils.

**Methodology:** Madhura Yeligeti.

**Project administration:** Hans Christian Gils.

**Resources:** Madhura Yeligeti, Hans Christian Gils.

**Software:** Madhura Yeligeti.

**Supervision:** Hans Christian Gils, Wolfgang Nowak.

**Visualization:** Madhura Yeligeti.

**Writing – original draft:** Madhura Yeligeti.

**Writing – review & editing:** Madhura Yeligeti, Hans Christian Gils, Wolfgang Nowak.

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
