## [Decision Letter · Decision Letter 0]

8 Apr 2025

PONE-D-24-59030A composite metric for evaluating system resilience with non-idealistic performance curvesPLOS ONE

Dear Dr. Yeligeti,

Thank you for submitting your manuscript to PLOS ONE. After careful consideration, we feel that it has merit but does not fully meet PLOS ONE’s publication criteria as it currently stands. Therefore, we invite you to submit a revised version of the manuscript that addresses the points raised during the review process.

We look forward to receiving your revised manuscript.

Kind regards,

Yuanchao Liu

Academic Editor

PLOS ONE

Journal Requirements:

“The research in this project was sponsored through the project 'ReMo-Digital' funded by the German Federal Ministry for Economic Affairs and Climate Action (BMWK) under grant number 03EI1020B, supporting the authors Madhura Yeligeti and Hans Christian Gils. The scientific contributions of Wolfgang Nowak are supported by the Stuttgart Center for Simulation Science (SimTech)”

Reviewers' comments:

Reviewer's Responses to Questions

**Comments to the Author**

1. Is the manuscript technically sound, and do the data support the conclusions?

Reviewer #1: Yes

Reviewer #2: Yes

2. Has the statistical analysis been performed appropriately and rigorously?

Reviewer #1: Yes

Reviewer #2: Yes

3. Have the authors made all data underlying the findings in their manuscript fully available?

Reviewer #1: Yes

Reviewer #2: No

4. Is the manuscript presented in an intelligible fashion and written in standard English?

Reviewer #1: Yes

Reviewer #2: Yes

5. Review Comments to the Author

Reviewer #1: Resilient system design is crucial in engineering, requiring quantitative resilience assessment through various metrics. Traditional metrics depict system responses to shocks as inverse bell-shaped, triangular, or trapezoidal curves, defining disruption and restoration phases. However, these models fail for irregular responses. This paper introduces a composite metric that integrates diverse system performance curves and a user-defined threshold. Validation through expert surveys confirms its effectiveness, and benchmarking highlights its adaptability across fields with non-ideal response behaviors.

Reviewer #2: I would recommend the following suggestions

I would recommend to format the manuscript according to the journal format/ guidelines. For that checkout recent Plos One Publications.

Line spacing and line numbers should be according to journal format.

In-text References and End bibliography should be rearranged accordingly as per journal's guidelines.

Figures and tables must be arranged as per journal's format.

Figures legends should be right after the paragraph they are discussed in.

Reformat paragraph alignment.

All supporting information should be provided in separate, Docx file.

6. PLOS authors have the option to publish the peer review history of their article (what does this mean?). If published, this will include your full peer review and any attached files.

Reviewer #1: **Yes: **Dhaneshwar Prasad Sahu

Reviewer #2: No

---

## [Author Response · Author response to Decision Letter 1]

18 May 2025

Thank you for the time and review. We have tried to address all the comments and they are explained point-by-point in the response-to-reviewers document. An additional note: we have modified our data avilability and now all data required for transparency and replication of this study are provided as supporting information without restrictions.

---

## [Decision Letter · Decision Letter 1]

20 Aug 2025

PONE-D-24-59030R1A composite metric for evaluating system resilience with non-idealistic performance curvesPLOS ONE

Dear Dr. Yeligeti,

Thank you for submitting your manuscript to PLOS ONE. After careful consideration, we feel that it has merit but does not fully meet PLOS ONE’s publication criteria as it currently stands. Therefore, we invite you to submit a revised version of the manuscript that addresses the points raised during the review process.

**Please revise accordingly**

We look forward to receiving your revised manuscript.

Kind regards,

Zhengmao Li

Academic Editor

PLOS ONE

**Journal Requirements:**

**Additional Editor Comments:**

please revise accordingly

Reviewers' comments:

Reviewer's Responses to Questions

**Comments to the Author**

1. If the authors have adequately addressed your comments raised in a previous round of review and you feel that this manuscript is now acceptable for publication, you may indicate that here to bypass the “Comments to the Author” section, enter your conflict of interest statement in the “Confidential to Editor” section, and submit your "Accept" recommendation.

Reviewer #1: All comments have been addressed

Reviewer #2: All comments have been addressed

Reviewer #3: (No Response)

Reviewer #4: (No Response)

Reviewer #5: All comments have been addressed

2. Is the manuscript technically sound, and do the data support the conclusions?

Reviewer #1: Yes

Reviewer #2: Yes

Reviewer #3: Yes

Reviewer #4: Yes

Reviewer #5: Yes

3. Has the statistical analysis been performed appropriately and rigorously?

Reviewer #1: Yes

Reviewer #2: Yes

Reviewer #3: Yes

Reviewer #4: Yes

Reviewer #5: Yes

4. Have the authors made all data underlying the findings in their manuscript fully available?

Reviewer #1: Yes

Reviewer #2: Yes

Reviewer #3: No

Reviewer #4: Yes

Reviewer #5: Yes

5. Is the manuscript presented in an intelligible fashion and written in standard English?

Reviewer #1: Yes

Reviewer #2: Yes

Reviewer #3: Yes

Reviewer #4: Yes

Reviewer #5: Yes

6. Review Comments to the Author

**Reviewer #1:** (No Response)

**Reviewer #2: **All of my comments have been addressed. Therefore, I suggest to accept this manuscript.

An amazing piece of work by authors. Keep it up and Good luck.

**Reviewer #3: **The authors explain that survey data are available upon request, this does not fully align with PLOS ONE’s data availability policy, which requires unrestricted public access to underlying data. The authors are encouraged to deposit anonymized survey data in a public repository or include it in the supplementary materials to fulfill this requirement.

**Reviewer #4: **The manuscript presents a novel composite resilience metric addressing limitations of existing methods, particularly for non-idealized system performance curves. The work is theoretically sound, methodologically rigorous, and well-supported by a survey and benchmarking. However, revisions are needed to enhance clarity, justify assumptions, and strengthen the discussion.

1-The composite metric fills a clear gap in resilience quantification for atypical performance curves. However, the introduction should better contextualize how this work advances beyond prior summaries.

2-The survey’s sample size (41 respondents) is modest. Discuss potential biases (e.g., dominance of energy system researchers) and generalizability.

3-Update citations to include 2023–2024 literature on resilience metrics (e.g., Raoufi et al. 2023).

4-Emphasize the metric’s adaptability to other domains (e.g., healthcare, transport) in the conclusion.



**Reviewer #5: **The author provided a satisfactory reply to my feedback, and based on that, they made thorough revisions to the article.

7. PLOS authors have the option to publish the peer review history of their article (what does this mean?). If published, this will include your full peer review and any attached files.

Reviewer #1: **Yes: **Dhaneshwar Prasad Sahu

Reviewer #2: No

Reviewer #3: No

Reviewer #4: No

Reviewer #5: No

---

## [Author Response · Author response to Decision Letter 2]

10 Oct 2025

Thank you to all reviewers for the feedback and support.

To reviewer #3 : We had made all data available already at submission.We appreciate the review and urge you to check our data availability statements again and let us know if something is still missing.

To reviewer #4 : thank you for your time and thoughts. We have accounted for them in our current revision. We anticipate that this addresses all your suggestions and look forward to a positive response.

---

## [Decision Letter · Decision Letter 2]

20 Oct 2025

A composite metric for evaluating system resilience with non-idealistic performance curves

PONE-D-24-59030R2

Dear Dr. Yeligeti,

We’re pleased to inform you that your manuscript has been judged scientifically suitable for publication and will be formally accepted for publication once it meets all outstanding technical requirements.

Kind regards,

Zhengmao Li

Academic Editor

PLOS ONE

Additional Editor Comments (optional):

Reviewers' comments:

Reviewer's Responses to Questions

**Comments to the Author**

1. If the authors have adequately addressed your comments raised in a previous round of review and you feel that this manuscript is now acceptable for publication, you may indicate that here to bypass the “Comments to the Author” section, enter your conflict of interest statement in the “Confidential to Editor” section, and submit your "Accept" recommendation.

Reviewer #4: All comments have been addressed

Reviewer #5: All comments have been addressed

2. Is the manuscript technically sound, and do the data support the conclusions?

Reviewer #4: Yes

Reviewer #5: (No Response)

3. Has the statistical analysis been performed appropriately and rigorously?

Reviewer #4: Yes

Reviewer #5: (No Response)

4. Have the authors made all data underlying the findings in their manuscript fully available?

Reviewer #4: Yes

Reviewer #5: (No Response)

5. Is the manuscript presented in an intelligible fashion and written in standard English?

Reviewer #4: Yes

Reviewer #5: (No Response)

6. Review Comments to the Author

Reviewer #4: The authors have provided excellent responses. Please thoroughly check the font size, lowercase first letter, expression, and other formatting issues throughout the article to ensure compliance with the journal's acceptance criteria.

Reviewer #5: The author provided a satisfactory reply to my feedback, and based on that, they made thorough revisions to the article.

7. PLOS authors have the option to publish the peer review history of their article (what does this mean?). If published, this will include your full peer review and any attached files.

Reviewer #4: No

Reviewer #5: No

---

## [Editor Report · Acceptance letter]

PONE-D-24-59030R2

PLOS ONE

Dear Dr. Yeligeti,

I'm pleased to inform you that your manuscript has been deemed suitable for publication in PLOS ONE. Congratulations! Your manuscript is now being handed over to our production team.

Kind regards,

on behalf of

Dr Zhengmao Li

Academic Editor

PLOS ONE